# Lacrosse Athletes Load and Recovery Monitoring: Comparison between Objective and Subjective Methods

**DOI:** 10.3390/ijerph17093329

**Published:** 2020-05-11

**Authors:** Richard Hauer, Antonio Tessitore, Reinhard Knaus, Harald Tschan

**Affiliations:** 1Centre for Sport Science and University Sports, University of Vienna, 1150 Vienna, Austria; a01065548@unet.univie.ac.at (R.K.); harald.tschan@univie.ac.at (H.T.); 2Movimento Umano e dello Sport, University of Rome “Foro Italico”, 00135 Rome, Italy; antonio.tessitore@uniroma4.it

**Keywords:** heart rate variability, rate of perceived exertion, short recovery and stress scale for sports, total quality recovery, training load

## Abstract

Both objective (OM) and subjective (SM) methods are used in athletic studies, regardless of sport type, to identify and analyze load and recovery status of athletes. As little information exists about the comparison of these two methodologies, the aim of this study is to compare and contrast information that defines the relationship between both methods. Twelve international male lacrosse athletes participated in this study over the course of which participants heart-rate-variability and questionnaire-data were collected. Statistical analysis was performed to evaluate changes over time and correlations between used methods. Comparison between baseline values and competition showed a reduction in root-mean-square of successive differences (RMSSD) (*p* < 0.01) and the proportion of beat-intervals (NN) that differ by more than 50 ms divided by total number of NNs (pNN50) (*p* < 0.01). Further, RMSSD values showed differences during competition with large effects (*p* = 0.02; η^2^ = 0.24). SM (*p* < 0.01) showed different progression during competition. Correlation was found for used SM and OM, when considered separately. No evidence for a reliable prediction of OM values using SM could be found. According to these findings, we recommend using a combination of SM and OM data to quantify the physiological stress of training and competition, respectively.

## 1. Introduction

An increase in long-term and peak performance at certain times are essential prerequisites for success in sports. To reach the targeted aims a structured variation in training volume and intensity is a fundamental planning paradigm to maximize performance in accordance with the athlete’s needs. Accordingly, improvement in performance is only possible if a sufficient quality and quantity of recovery is provided [1]. In this context, the assessment of the load and recovery status play an ever important role [2,3]. The monitoring of load and recovery in training and competition has been attracting increasing attention in recent years due in no small part to continuing technological developments and availability of monitoring devices [4,5]. In the same manner, findings by Cardinale and Varley [4] show that the assessment of training load (TL) across different sports has increased as well. Tracking athletes’ load and recovery status provides a better understanding of training, training loads optimization, and individually structured program design to both improve performance and reduce injuries [6]. Foster et al. [5] see a tendency to measure two different aspects in training. Firstly, the external load (EL) measurement, such as total distance, distance in different speed zones covered, and number of acceleration and deceleration are used for training observations. Secondly, internal load (IL) measurement, which includes heart rate (HR), time spent in different heart rate zones (HRz) and athletes’ perception about load and recovery are evaluated. Subsequently, the EL and IL relationship and the effect of their interaction on sport performance has to be observed [5]. In this context, Fox et al. [7] speak of a dose-response relationship which consists of a set EL that determines the training intensity and an individual physiological reaction. For a better understanding of the dose-response relationship a wide array of methods exist, but few of these have shown scientific reliability and validity [8].

A common tool to assess athletes’ subjective perception is the questionnaire [1,8,9]. The often- used “rate of perceived exertion” (RPE) questionnaire can be used to assess subjective load status. The RPE seems to correlate with the more objective HR values and can therefore be used to asses athletes IL [8]. Similarly, the “total quality recovery” (TQR) questionnaire can be used to assess the recovery status. Utilizing the 6–20 BORG-scale [10], RPE and TQR provide an easy to use and comparable method to assess athletes’ load and recovery status [9]. In the same manner, Hitzschke et al. [1] designed the short recovery and stress scale for sports (SRS) questionnaire, which can be used to assess an athletes’ subjective perception. Further, for a more objective method of training control researchers and practitioners often use the heart-rate-variability (HRV) [11,12]. Even with some diagnostic difficulties in interpretation the short-term HRV measurement seems to be a reliable and valid method to assess athletes load and recovery status when used properly [8,13]. In this context, the most commonly used HRV parameters are the “root-mean-square of successive differences between normal heartbeats” (RMSSD) and the “proportion of the number of pairs of successive- intervals that differ by more than 50 ms divided by the total number of intervals” (pNN50) [14,15].

Other frequently used method to assess neuromuscular fatigue are the counter-movement-jump (CMJ) [16] or the more expensive measurement and comparison of baseline biomarkers such as testosterone-cortisol ratios. This type of information can be used to assess an individual’s break down level of homeostasis, which can then give information about recovery status [17]. Even saliva measurements seem to be a reliable and valid method, but practically can be limited due to the cost and availability of equipment and resources.

In summary, a variety of different methods to monitor the load and recovery status of athletes exist. Each of these methods brings advantages and disadvantages with respect to cost and practicality. Overall, the monitoring process can be categorized in three different groups. Exercise measurements can be used to give information about the load athletes are exposed to during physical activity, post-exercise measurements can be used to assess the individual response to load and resting measurements can give information about the recovery status and readiness of athletes over the course of training and competition [18,19,20].

As most studies usually focus on one training load and recovery assessment method to assess training load and recovery status limited studies exists, which investigate the relationship between OM and SM. While the results from some studies have indicated a relationship between training load and recovery assessment methods, a meta-analysis by Zahn et al. [21] shows inconsistent correlation values in the relevant literature. Further research is needed to investigate and identify the relationship and parallels of different monitoring methods. To the best of the authors knowledge no study about load and recovery status in the sport of lacrosse have been done so far. In the same manner, no data about possible OM and SM to state athletes load and recovery status over the course of competition exists. Furthermore, with little research about the activity profile in lacrosse match-play and training [22,23,24,25,26] it seems to be important to provide information of athletes’ response to demands over the course of competition in the sport of lacrosse.

Based on the literature mentioned above, the aim of this study was to give information about practical methods for resting measurements to evaluate athletes’ load and recovery status over the course of competition. Furthermore, results of this study should provide coaches, researchers, and practitioners working in the field of lacrosse with a better understanding of the use of OM and SM to evaluate athletes load and recovery status over the course of competition. In order to achieve this goal, the research group focused on HRV measurements and short questionnaires. Firstly, these methods seem to be relatively easy to implement and do not require expensive equipment or resources. A current research has proven it a reliable and valid method in relationship to expenditures. Secondly, HRV and questionnaires can be used on the road and are both user-friendly and comprehensive, which seem to be crucial for an athlete’s acceptance. Aside from the applicability, as the main goal was to give information about the relationship between OM and SM, the research group categorized HRV measures as an objective method and questionnaires were chosen as the subjective method to assess load and recovery status over the course of competition. In accordance to findings in recent literature [1,20] we hypothesized that each method by itself show changes in athletes load and recovery status over the course of competition. Furthermore, it was hypothesized that both monitoring methods show similar results and therefore a relationship between OM and SM exists. If this is the case, it can then be concluded that only one method needs to be practically implemented by coaches and practitioners to evaluate the load and recovery status of their athletes.

## 2. Materials and Methods

### 2.1. Ethical Approval

The study was approved by the University of Vienna Ethics Review Board (Reference Number: 00190) and was conducted in accordance with the Declaration of Helsinki. All subjects provided informed consent after reading a description of all research procedures and received guarantees on data anonymization.

### 2.2. Subjects

Twelve male lacrosse athletes of the Austrian national team, which competed at the European Indoor Lacrosse Championships (EILC), participated in this study. Subjects were aged between 20 and 36 years with a mean of 26.8 ± 5.6. Athletes’ anthropometric data showed a mean height of 178.4 ± 5.1 cm and a body mass of 78.7 ± 8.8 kg. The calculated body-mass-index (BMI) ranged from 19.0–28.4 with a mean of 24.7 ± 2.4. Only data of subjects that participated in at least one game were included in the evaluation. Furthermore, for a more homogeneously collective sample, the goalkeepers were excluded as the demands of this position differ significantly compared to other positions and players.

### 2.3. Procedure

During the twelve days prior to the EILC the athletes’ baseline HRV was evaluated using a Polar RS800 heart rate belt and clock device (Polar Electro, Kempele, Finland). The athletes were given verbal instructions on the use of the equipment, an instruction manual, and detailed guidance on the measurement schedule. Prior to, and during the eight days of competition, the athletes monitored their HRV values every morning. HRV data consisted of RMSSD and pNN50. To evaluate the relevant data, subjects were instructed to put on the device and lie on the bed for 5 min before getting up in the morning. All tracked HRV data was stored on the athletes’ allocated Polar RS800 clock. In a second step, data was transferred to a computer and analyzed using Kubios HRV Standard v3.0.1 (Kubios Ltd, Kuopio, Finland) by the research group. Next all data was collected and summarized, and a “Masterfile” for statistical analyzation was created in Microsoft Excel (Microsoft Corporation, Redmond, WA, USA). Data included RMSSD and pNN50 daily baseline values, baseline mean values, values for each day of competition and mean values over the course of competition for each player. In a final step, the Excel “Masterfile” was transferred to a statistical program for statistical analyzation. RMSSD and pNN50 values were categorized as OM for all further analysis and results.

In addition to HRV measurement athletes recorded their subjective perception during the competition period. Therefore, athletes’ completed Short Recovery and Stress Scale for Sports (SRS) questionnaire [1]. SRS consisted of 4 questions including muscular stress, activity status, emotional balance, and overall demands/condition to determine stress level (SRS-S):Overall demands/condition. E.g., tired, overloaded, strengthlessstrongly disagree—0 → 1 → 2 → 3 → 4 → 5 → 6—strongly agree;

In the same manner, recovery level was asked by 4 questions including physical capacity, mental capacity, emotional balance, and overall recovery condition (SRS-R):Overall recovery condition. E.g., physically relaxed, recovered, restedstrongly disagree—0 → 1 → 2 → 3 → 4 → 5 → 6—strongly agree;

Athletes had to rate each question on a seven-point scale from 0 (strongly disagree) to 6 (strongly agree). Further, recovery quality was evaluated using the total quality recovery (TQR) questionnaire. Additionally, the rate of perceived exertion (RPE) questionnaire was used in accordance to Borg [10]. For a better comparison the 6 (very, very poor recovery) to 20 (very, very good recovery) Borg scale was used for TQR and RPE [9]. Athletes completed SRS and TQR questionnaires every day at the morning team meeting. RPE was registered individually 15 min after the end of each game. All tracked questionnaires were collected by the research group and data was entered into a Microsoft Excel file. In a second step the data was transferred to the “Masterfile”. Data included a point score for each day of competition and mean values over the course of competition for each player and each test performed. SRS, TQR, and RPE values were categorized as SM for all further analysis and results.

Competition consisted of seven games played over a period of eight days. The competition started Saturday and ended on the following Saturday. The Austrian team played six games in a row before they had a day off on Friday (day 7). This rest day consisted of two light training sessions, one in the morning and one in the afternoon, with a duration of 60 min each. On the eight day the final placement game took place. All games were played at different times between 8:00 am and 10:00 pm. A more detailed information of the schedule and score for each game played by Team Austria is provided in Table 1. All games were played on an indoor ice rink on a concrete surface (no ice). The regulation playing time of a game was 60 min in total, divided into four 15 min quarters, with 2-min quarter breaks and a 12-min half time break. Physiological demands like IL and EL and other influencing factors like sleep and nutrition, were not registered during and over the course of competition.

### 2.4. Statistical Analysis

All statistical analysis was performed with the software SPSS v24.0 (SPSS Inc., Chicago, IL, USA). Assumptions of normality were verified by Shapiro Wilk test and histograms. To determine changes over time for each of the independent variables separate repeated measurement of ANOVAS were performed. To evaluate differences between first and last day of competition a paired *t*-test was performed. Pearson’s Product-Moment Correlation was used to understand the relationship between objective and subjective methods. For prediction of metric data, multiple linear regression analysis was used. Effect sizes (ES) were calculated as partial eta-squared (η^2^) categorized as small = 0.01, moderate = 0.10, and large = 0.25. Additionally, effect sizes d according to Cohen et al. [27] were calculated. The magnitude of the inferences was determined as small (d = 0.2–0.5), medium (d > 0.5–0.8), large (d > 0.8–1.3), and very large (d > 1.3). All parameters are presented as mean ± standard deviation (SD) unless stated otherwise. The significance level for all tests was set at *p* ≤ 0.05. All calculations are based on a 95% confidence interval (CI).

## 3. Results

### 3.1. HRV Analysis

The mean base line HRV values, evaluated during the 12-day period before competition, were 77.79 ± 25.79 ms for RMSSD and 34.99 ± 14.61% for pNN50, respectively. Compared to the reported summarized mean values over the course of the eight days of competition (RMSSD: 64.08 ± 25.74 ms, pNN50: 28.11 ± 16.02%) a significant decrease in both values (RMSSD: *p* = 0.003, d = 1.17; pNN50: *p* = 0.001, d = 1.45) could be found (Table 2). Further, RMSSD showed differences with moderate ES over the course of competition on a daily base (F_(7,56)_ = 2.544; *p* = 0.024; η^2^ = 0.24). On the other hand, no differences were found for pNN50 (F_(7,56)_ = 2.088; *p* = 0.060; η^2^ = 0.21), as shown in Figure 1. Similarly, comparison between the first and last competition day did not show any differences in both HRV values (Table 2).

### 3.2. Questionnaires Analysis

As shown in Table 2, SM comparison between the first and last day of competition did show differences (*p* ≤ 0.021) with medium to large ES (d ≥ 0.78) for SRS and TQR values. RPE did not show differences (*p* = 0.054) but a tendency with medium ES (d = 0.62). Further, analysis regarding changes in SM values over the course of competition on a daily base did show similar findings. SRS-R (F_(7,77)_ = 4.438; *p* = 0.001; η^2^ = 0.29) and TQR (F_(7,77)_ = 4.433; *p* = 0.001; η^2^ = 0.29) fluctuated with a continuously decreased over the course of competition. In the same manner, SRS-S (F_(7,77)_ = 4.223; *p* = 0.001; η^2^ = 0.28) increased (Figure 2). Post hoc analysis did show differences in SRS-R first day and fifth day of competition (*p* = 0.004), and in TQR first to, third (*p* = 0.004), fourth (*p* = 0.032), fifth (*p* = 0.005), seventh (*p* = 0.020), and eighth day (*p* < 0.001). No significant differences but a strong trend with large ES were found between RPE values (F_(7,42)_ = 2.176; *p* = 0.056; η^2^ = 0.27).

### 3.3. Relationship between Subjective and Objective Monitoring Methods

Relationship analysis for mean value over the course of competition of all used methods are shown in Table 3. Correlations between RPE and SRS (SRS-S: *p* = 0.034, r = −0.613; SRS-R: *p* = 0.025, r = 0.639, respectively) were found. Similarly, TQR showed correlation with SRS-S (*p* = 0.044, r = −0.588) and SRS-R (*p* = 0.009, r = 0.717). Furthermore, SRS-S and SRS-R showed a high negative correlation (*p* = 0.042, r = −0.594). Additionally, analysis for RMSSD and pNN50 showed a high positive correlation (*p* < 0.001, r = 0.935).

Multiple linear regression for SM to predict the mean value over the course of competition for RMSSD (SRS-S: *p* = 0.673, β = −0.202; SRS-R: *p* = 0.599, β = −0.305; TQR: *p* = 0.248, β = 0.627; RPE: *p* = 0.825, β = −0.107, respectively) did not show the possibility for a reliable prediction (Table 4). Similarly, no reliable prediction for pNN50 for any of the SM (SRS-S: *p* = 0.953, β = −0.029; SRS-R: *p* = 0.390, β = −0.528; TQR: *p* = 0.229, β = 0,681; RPE: *p* = 0.678, β = 0.210, respectively) were found.

## 4. Discussion

Subjective and objective methods to evaluate an athlete’s load and recovery status are a common tool used by coaches and practitioners [7,28]. While on their own each method appears both reliable and valid, their relationship and comparability are still unclear and there has been a lack of data in team sports competition to give information about the usability of these methods. Therefore, the aim of this study was to (a) explore the usability of each method by itself and (b) investigate the relationship between SM and OM in the course of major international lacrosse competition. To our knowledge, this study is the first exploring load and recovery status of lacrosse athletes over the course of competition. Findings will improve the knowledge and awareness on how these methods can be used by coaches and practitioners.

Differences with large to very large ES between baseline and HRV values during competition could be found for RMSSD (d = 1.17) and pNN50 (d = 1.45), respectively (Table 2). The observed reduction in both values during competition is in accordance with findings by Bürklein et al. [29] who stated a reduction in RMSSD and pNN50 on consecutive days of sport activity. The comparison between first and last day of competition did not show differences in HRV values. These findings are in agreement with results by Egan-Shuttler et al. [30] in high school rowers. Nevertheless, the comparison of only two certain points can have a distorting effect. Therefore, a more detailed observation between each day over the course of competition was conducted. Data showed differences with moderate ES for RMSSD (η^2^ = 0.21) and pNN50 (η^2^ = 0.24), respectively. As shown in Figure 1 a tendency of differences between the first (*p* = 0.105) and second day (*p* = 0.106) of competition compared to the sixth day of competition could be found for RMSSD. No daily differences were observed in pNN50 values. One explanation for these different findings, between first and last vs. daily comparison, could be the rest day on day seven. This rest day could have caused a better recovery status and can therefore be an explanation for the similar HRV values compared to day one. Apart from that, daily comparison indicates a successive reduction when competitive games are played on a daily base. This can be seen in Figure 1 as HRV values show a continuous drop from day two to six. Again, the difference between day seven and eight to day one could be due to the tournament schedule, as the game on day six was played in the morning and athletes had a longer recovery phase until the final placement game on day eight. Nevertheless, these findings indicate that a rest day can give players enough time to recover after several days of competition. This should be considered by organizers when scheduling major events, as better recovery status could probably reduce participants injury risk. Furthermore, this may also have implications for coaches and professionals to appropriately plan training and consider the impact of upcoming tournaments.

Data of questionnaires values showed differences between first and last day of competition with medium to very large ES (SRS-S: d = 0.78; SRS-R: d = 0.91; TQR: d = 2.19; RPE: d = 0.62). The detailed results of a reduced recovery and increased load status are shown in Table 2 and Figure 2. The observed results in SRS are in common with findings by Hitzschke et al. [1] in elite hockey players. On reason for the different finding compared to HRV values could be that the evaluation of recovery status using questionnaires is not as precise and sensitive as the physiological response. Another influencing factor could have been the time lag between games and survey date. Nevertheless, a trend of adaption to first day values can be seen when looking at daily differences. The comparison between each day over the course of competition did show differences in SRS-R between first and fifth day (*p* = 0.04) and TQR between first and third (*p* = 0.04), fourth (*p* = 0.03), fifth (*p* = 0.01), seventh (*p* = 0.02), and eighth (*p* < 0.01) day (Figure 2). The trend to a reduction in subjective recovery status, with a raise in stress level values show similarities to HRV measurement. These findings would suggest that a relationship between SM and OM do exist.

Conversely, results of the inter-method relationship analysis between OM and SM did not show any correlations (Table 3 and Table 4). These results do indicate that no reliable prediction of RMSSD nor pNN50 can be made by using short stress and recovery level questionnaires. Even though this might be true, another reason for this outcome could be the design and methods used in this study. It is possible that the used methods for OM (RMSSD and pNN50) and/or SM (SRS, TQR, RPE) are not sufficiently precise for reliable and comparable load and recovery status analyzation over the course of competition. However, intra-method relationship analysis did show correlations for the used methods (Table 3). In this context, TQR and RPE values showed a positive coincidence with SRS-R and a negative coincidence with SRS-S values. These findings indicate that athletes with higher load status tended to have better recovery status on the following day. Contrarily, higher SRS-S values in the morning showed lower RPE values after the game. Additionally, a high negative correlation was found for SRS-S and SRS-R. This is unsurprising, as a high stress level over a time leads to a lower recovery status. Similarly, RMSSD and pNN50 values did show a high positive correlation, which indicates that both used methods show similar results in monitoring athletes load and recovery status. Therefore, it might be possible that less SM and OM data than used in this study is needed to get reliable und enough information about the load and recovery status over the course of competition.

Overall, the results of this study support findings of previous studies that indicate that OM and SM are appropriate tools for resting measurement of load and recovery status of athletes and that on their own each appears valid and reliable [1,28,31,32]. Further, this study showed that these methods seem to be appropriate for the sport of lacrosse. HRV measurement was used as an OM for athlete monitoring which proved to be a practical choice even considering the special equipment, qualified personnel, and the time for instruction and measurements that are needed [33]. While, questionnaires were used as a SM and required less expenditure and instruction to assess data, subjective evaluation can cause higher error rates especially when individuals are not familiar with them. Therefore, it is recommended that athletes should be familiarized with the questionnaires prior to use, to reduce incorrect interpretation.

Results of this study showed intra-method correlations. The high positive correlation of RMSSD and pNN50 indicate that one parameter seems to be enough to give information of the parasympathetic activity, degree of relaxation, and the recovery ability of the organism. In the same manner, TQR and RPE showed moderate to high correlations with SRS data. Like OM findings, SM indicate that either TQR in combination with RPE or SRS questionnaires seem to be efficient to give information about athletes load and recovery status over the course of competition. However, based on our results, a prediction or transferability from one method (OM–SM) to the other seems to be questionable. A reliable prediction of RMSSD and pNN50 using short load and recovery questionnaires does not seem feasible. Based on the results, the authors conclude that SM should not be used to replace OM. Rather both methods should be implemented and used to reduce errors and incorrect interpretation.

Findings of this study must be interpreted with caution concerning the small number of athletes and games observed. Further, match demands such as EL and IL were not recorded in this study. Varied individual responses to match load based on an individual athlete’s fitness level and health, as well as playing time on the field, can be influencing factors. Additionally, physical demands can vary widely depending on the course of game play. Moreover, other non-match relevant factors can influence athlete’s recovery status such as quality and amount of sleep, nutrition, state of mind, and morale. As neither match demands nor non-match related influencing factors were evaluated in this study, outcomes and interpretations should be critically examined. Another limitation to the present study is the use of HRV values as an objective monitoring method. While recent literature state this method as reliable and valid many other options do exist to evaluate load and recovery status of athletes over the course of competition. Results regarding predictability and the correlations described can be caused by the methods used in this study. It is therefore possible that the use of other methods would have shown contradictory results [34,35]. With these limitations in mind, future studies should use the knowledge gained from this study to (a) develop new and better study designs on a critical base; (b) use similar methods and study designs to include more participants and data points; and (c) observe the outcomes of different methods than used in this study and give more information about the usability and predictability of OM and SM in team sports during training and competition.

## 5. Conclusions

In accordance to the findings of this study, it can be concluded that objective and subjective methods are two independent methods to evaluate load and recovery status during competition. Each method on its own seems to give valid and reliable information, but do not allow any direct conclusion to be drawn from one method to the other, respectively. On the other hand, different observed SM showed correlations between each other. Therefore, we conclude that RPE in combination with TQR or SRS only are enough to give subjective information about athletes’ load and recovery status. Nevertheless, the authors recommend coaches and practitioners to combine OM and SM to evaluate athletes’ load and recovery status. One reason for this conclusion is the missing relationship and reliable prediction of OM using SM found in this study. Furthermore, each method of itself shows advantages and disadvantages. SM are not as objective but give a fast and direct response as they are easy to implement and do not require expensive or additional equipment. OM on the other hand seem to provide a dispassionate and more reliable information but require additional equipment and time. Taking this into account, the authors recommend using the two methods for reciprocal control of each other, to limit incorrect interpretation and improve explanatory power.

## Figures and Tables

**Figure 1 ijerph-17-03329-f001:**
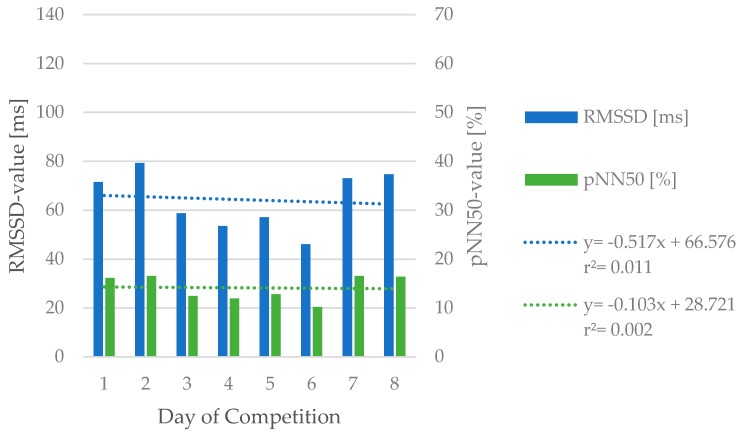
Daily differences for RMSSD and pNN50 over the course of competition. Results are presented as mean ± SD (95% CI) in (ms) for RMSSD and (%) for pNN50.

**Figure 2 ijerph-17-03329-f002:**
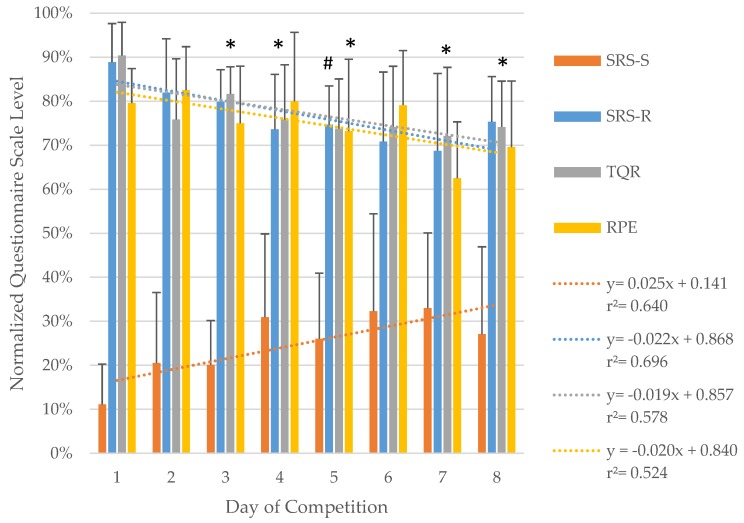
Daily differences for SRS-S, SRS-R, TQR, and RPE over the course of competition. Results are presented as mean ± SD (95% CI) of normalized questionnaires scale level; ^#^—Significant difference for SRS-R to 1st day of competition; *—Significant difference for TQR to 1st day of competition.

**Table 1 ijerph-17-03329-t001:** Austrian National Team competition schedule at the European Indoor Lacrosse Championships.

Date	Start-Time	Team-1	Team-2	Score
Saturday, 8 July	15:15	Austria	Sweden	12:8
Sunday, 9 July	11:00	Netherlands	Austria	10:19
Monday, 10 July	15:15	Switzerland	Austria	11:10
Tuesday, 11 July	20:00	Austria	Poland	9:8
Wednesday, 12 July	17:45	England	Austria	18:8
Thursday, 13 July	13:15	Sweden	Austria	13:8
Friday, 14 July	Day off			
Saturday, 15 July	10:15	Austria	Netherlands	20:3

**Table 2 ijerph-17-03329-t002:** Differences between objective and subjective monitoring methods during competition. Results are presented as mean ± SD (95% CI).

Monitoring Method	Mean	±SD	*p-*Value	d
RMSSD-pre competition	77.79	25.97	0.003 **	1.17
RMSSD-during competition	64.08	25.74		
pNN50-pre competition	34.99	14.61	0.001 **	1.45
pNN50-during competition	28.11	16.02		
RMSSD 1st day of competition	71.48	22.97	0.603	0.17
RMSSD last day of competition	76.55	43.92		
pNN50 1st day of competition	32.33	15.02	0.929	0.03
pNN50 last day of competition	32.65	21.39		
SRS-S 1st day of competition	2.67	2.19	0.021 *	0.78
SRS-S last day of competition	6.50	4.76		
SRS-R 1st day of competition	21.23	2.10	0.009 **	0.91
SRS-R last day of competition	18.08	2.47		
TQR 1st day of competition	18.08	1.51	0.000 **	2.19
TQR last day of competition	14.83	2.08		
RPE 1st day of competition	15.92	1.56	0.054	0.62
RPE last day of competition	13.92	3.00		

Significant difference: * *p* ≤ 0.05; ** *p* ≤ 0.01. SD—standard deviation; d—magnitude of the inferences.

**Table 3 ijerph-17-03329-t003:** Pearson product correlation of mean values over the course of competition for all used methods.

Monitoring Method	RPE	RMSSD	pNN50	SRS-R	SRS-S	TQR
RPE	1.000	0.056	0.149	0.639 *	−0.613 *	0.370
RMSSD	0.056	1.000	0.935 **	0.211	−0.326	0.487
pNN50	0.149	0.935 **	1.000	0.120	−0.246	0.397
SRS-R	0.639 *	0.211	0.120	1.000	−0.594 *	0.717 **
SRS-S	−0.613 *	−0.326	−0.246	−0.594 *	1.000	−0.588 *
TQR	0.370	0.487	0.397	0.717 **	−0.588 *	1.000

Significant difference: * *p* ≤ 0.05; ** *p* ≤ 0.01. Rate of perceived exertion (RPE); Root-mean-square of successive differences between normal heartbeats (RMSSD); Proportion of the number of pairs of successive- intervals that differ by more than 50 ms divided by the total number of intervals pNN50; Short recovery and stress scale for sports (SRS-R for recovery and SRS-S for stress level); Total quality recovery (TQR).

**Table 4 ijerph-17-03329-t004:** Prediction analysis of subjective monitoring methods for RMSSD and pNN50 mean value over the course of competition.

Predictor	RMSSD			pNN50		
	β	t	*p*	β	t	*p*
SRS-S	−0.202	−0.441	0.673	−0.029	−0.061	0.953
SRS-R	−0.305	−0.551	0.599	−0.528	−0.917	0.390
TQR	0.627	1.261	0.248	0.681	1.317	0.229
RPE	−0.107	−0.229	0.825	0.210	0.433	0.678

Note: (β) Standardized coefficient; (t) difference relative to the variation; (*p*) probability value.

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
