# Peer review of "Lacrosse Athletes Load and Recovery Monitoring: Comparison between Objective and Subjective Methods"

_ijerph, 2020, doi:10.3390/ijerph17093329_

Round 1

Reviewer 1 Report

The authors should be commended on their work to improve the overall quality and clarity of the manuscript. I have some additional comments and suggestions which can be found below.

Abstract L21-22: P cannot equal 0.00 instead this should be presented as p<0.001. This also occurs in other sections of the manuscript. Please check carefully.

Intro L33: 'An increase...'

L148: 'which investigate...'

L153: 'Based on the literature mentioned above...'

At least some discussion of why this study is important in Lacrosse should be presented in the introduction to give the reader context and to provide importance of the research question to professionals working in this sport.

L181: 'collective sample'

L277: Here it would be good to provide information about the time of day each competition occurred at. This may help the reader make more sense of the daily fluctuations observed in the data as it may be related to the time since the previous match.

L285: 'each of the independent variables separate repeated..'

Figure 1 an 2 top panels: equation is not presented.

L335: remove the term 'strong'

Figure 2. * and or # should be used to denote significance for each variable.

L403: 'Time to recover'

L404-405: This may also have implications for coaches and professionals to appropriately plan training and consider the impact of upcoming tournaments.

L412: 'Physiological response'

L420-422: The wording here does not read well.

L429: remove 'did'

L530-534: Amend to proper paragraph format.

L592: 'objective' is just repeating the type of measure consider using another word.

Author Response

Thank you for the suggestions to help to improve the quality of the paper. See the attachment for a step by step Information about the made changes according to your statements. 

Best Regards

Reviewer 2 Report

The authors have done a fine job in addressing my original concerns regarding this manuscript.

Author Response

Thank you for your help to improve the quality of the paper and the positive feedback.

Best Regards 

This manuscript is a resubmission of an earlier submission. The following is a list of the peer review reports and author responses from that submission.

Round 1

Reviewer 1 Report

This study compared objective and subjective measures of load and recovery monitoring in lacrosse athletes. Data was obtained from 12 male athletes over eight days of competition. Collectively, the authors conclude that using a combination of objective (e.g. heart rate variability) and subjective (e.g. rating of perceived exertion) is recommended. Unfortunately, there are major concerns with the clarity of the study design and methods, presentation of the narrative in the introduction and discussion and amendments required to tables and figures. At this stage I believe the manuscript is not suitable for publication in its current format. I have provided several general and more specific comments below.

Specific comments:

Note, many writing errors are presented. Some but not all of these are listed below.

Line 13: ‘relationship’

Line 18: RMSSD and pNN50 need to be defined here.

Line 50: ‘Perceived exertion’

Line 54: error for ‘the by’?

Line 57: ‘researchers’

Line 58: ‘diagnostic difficulties’

Lie 65: ‘status limited studies..’

Line 71: what is meant by show value changes here?

Line 144: ‘Further, analysis regarding…’

Line 157: ‘showed a correlation…’ and ‘with SRS’.

Line 169: ‘method by itself’

Line 170: ‘over the course’

Remove title from Figure 1 and 2.

Figure 2 should show significance points as outlined in line 148.

Table 2 caption: significance values are noted but none are presented in the actual table. Either add as necessary or remove this from the caption.

Linear trendlines offer no insight as the r value and equation of the line is not presented in relation to the data.

Row three and four of Table 1 seem to have the same sub-heading but different values.

Line 213-214: This statement is unclear.

General comments:

One of the major limitations of the study is that very little information is presented about the tournament/competition the athletes participated in. For example, there is no data explaining the load undertaken during each match, or any relationship between match load and the responses observed (objective and subjective) for each individual. There are also many other factors outside of the match that can influence recovery such as sleep and nutrition. Were any of these recorded? At the very least discussion of these limitations is required.

How many matches were completed for each player over the 8 day period? Did all of these occur at the same time of day? What about effects of weather conditions on load and recovery outcomes?

Line 113: Assume separate ANOVAS were conducted for each outcome variable. Please amend.

For small sample sizes Hedge’s G effect size is preferred.

The discussion reports most of the results but is limited by interpretation of the findings. In particular, lines 186-197 lack narrative and insight by the authors into the reasons that these results may have occurred.

Lastly, apart from the specific writing edits and comments provided above, the general flow and wording in most of the sections requires significant amendment above what can be provided as specific point-by-point responses listed here. Therefore, I think significant improvement is required to the manuscript.

Reviewer 3 Report

This study is addressing an interesting issue: the (complementary) nature of subjective (SM) and objective methods (OM) to assess the load and recovery of lacrosse players during competition. Subjective and objective methods are widely used in practice and especially the combination of both opens a lot of interesting opportunities. Currently, the numbers of methods and specific variables used for monitoring (load and recovery) are very large. The comparison between subjective and objective methods is therefore relevant for both scholars and practitioners. This could benefit further studies investigating how to monitor and assess athletes in different sports than lacrosse and to expand the knowledge on athletic surveillance and individual assessing methods

Although I’m pleased to see this topic addressed in this paper and the paper uses clear language, I do have some serious concern about this paper. I will list them up bullet wise.

  • I miss a connection with the aims and the scope of this journal. It is important (in order to have it published in this paper) to make a clear connection in both the introduction and the conclusion. At the moment this is a paper which probably would fit better in a sports related journal focussing on physiology.
  • I belief that the ‘introduction’ section is very limited and lacks a thorough review of literature regarding the topics presented in the paper. Moreover, the introduction does not pay sufficient attention to the subject under consideration and the methods chosen in the study. In the first part of the introduction, for example, several topics are raised that are related to each other, but the manuscript does not really give an in-depth overview.
  • In the introduction lots of possible measures and methods are described. Yet, there are also lots of other possibilities that are not described. How did you make this selection?
  • In the introduction section, external (EL) and internal load (IL) are mentioned as relevant measures to assess the physical load of athletes. Yet, these measures are not described or mentioned in the method/procedure. I would like to see the integration of EL and IL into assessment procedure.
  • The purpose of the study should be formulated more specifically (and this should also be clear in the rest of the paper). The title claims to focus on the comparison of methods, but that is not really seen in the results section. DO you have support (from literature) for your hypotheses?
  • It is not clear why and how you selected your measurements. How were they selected and operationalized? The paper gives the impression that this was done rather random. A thorough analyses based on review could have helped here. Why do the authors choose HRV over other possible measures? I propose to frame the chosen measures in a theoretical model, some of which are discussed in the introduction, and to use them to identify and justify the chosen methods and variables. These variables should therefore be included in the aim of the study. Only using OM and SM is very generic. What would be the difference between performance and peak performance? (Line 28) What is subjective monitoring, is perception not always subjective? (line 49). What ‘training methods’ are referred to in line 72?
  • I can imagine a link from HRV about stress to stress and recovery, but the authors don't make this clear/explicit? Also, with regard to the discussion: I miss a critical approach to the choice of HRV (I have some doubts...).
  • With regard to the method section I also have some major concerns. The variables chosen are explained quit well, but it was not clear to me which variable was used for OM and which variables for SM. In comparing these methods that would be expected to be clearly stated in this section of the paper. The authors want to address ‘Load and recovery status’ both with OM and SM measures as stated in the aim. Within the procedures it is not clear which variables fit which type of status and to which type of monitoring it is fitted. I feel there is a need to do that. This is unclear.
  • I believe it is also important to present an overview of how the questionnaires were implemented in the study. How many times were the athletes asked to provide answers to the questionnaires such as SRS? Did they do it only at the end of the study, during the study or several times?
  • In line with this, the ‘result’ section is very limited. Please provide more information, more results. Moreover, following the previous remarks about the introduction and methods sections, I believe the results section should be restructured. The authors claim that the comparison of methods is the aim of this study. Following this, I would expect to see a strong focus on this in the results section. Those results (paragraph 3.3) get quit little attention in the paper. With regard to that paragraph I have some questions: Which data did you use for these correlations? Did the authors used day data (separately) or everyday data (combined)? Are correlations done for the linear relationship over time?
  • In the results section tables and figures are shown. What do they tell us. Averages are shown, but it may be more interesting to focus on individual patterns of athletes and comparisons between the two ways of measuring.
  • Figure 1 shows that HRV measures are the OM. To what factor do the authors apply these objective monitoring, Load or Recovery? Did the authors account for competition days and non-competition days? When were these competition days? I also have trouble with understanding figure 2. As far as I understand, data are used (which have a different unit of measure), but the y-axis has the same absolute dimensions for all results. I feel it should be helpful to plot normalized data instead. Providing all p-values would support good interpretations.
  • I believe the discussion could benefit from more focus, from deleting repetitive material and providing a stronger integration with literature. The discussion section doe not provide sufficient deeper interpretation and discussion of the results. For example, information about the competition schedule could help in interpreting the results deeper and with a more practical approach.
  • I would suggest to restructure the discussion. I would suggest taking the first part of the discussion to the introduction. It is more specific and clearly defined than the aim of the study in the introduction. Still I believe OM and SM should be justified more.
  • In the discussion section the authors state that they explored usability. Is this usability what they studied.? I would rephrase this.
  • In lines 200-202 the authors make a rather strong claim. Do you really have hard evidence for this?
  • While I understand that the focus of the document is on providing information on the applicability of OM and SM to evaluate the workload and recovery status of athletes during competition, I believe it is relevant to identify possible directions and implementations to combine OM and SM for athletic monitoring.
  • I also miss a discussion of limitations and a critical reflection to the chosen approach. How would the authors suggest using these methods for predictability? (line 208) and to what status or other training of competition factor could this be used according to your results? For the conclusion I seriously doubt whether HRV measures could give any insight in load. I question whether HRV measures on itself are really reliable as put forward in the conclusion (also given the fact that the athletes in this study did self-monitoring).

Some minor comments:

  • The abstract needs rewriting to be understood better. Lots of abbreviations are used that doesn’t make sense reading the abstract only. Avoid using terms such as RMSSD and pNN50 in the abstract without giving further explanation.
  • Athletes’ is used incorrectly in several occasions, for example line 99.

I wish the authors luck in the further development of this paper. The paper is addressing an interesting topic, but I believe quite some additional effort is needed.